# Anticancer Activity of Novel Difluorinated Curcumin Analog and Its Inclusion Complex with 2-Hydroxypropyl-β-Cyclodextrin against Pancreatic Cancer

**DOI:** 10.3390/ijms24076336

**Published:** 2023-03-28

**Authors:** Sangita Bhattacharyya, Hindole Ghosh, Obdulia Covarrubias-Zambrano, Krishan Jain, K. Venkateswara Swamy, Anup Kasi, Ameer Hamza, Shrikant Anant, Michael VanSaun, Scott J. Weir, Stefan H. Bossmann, Subhash B. Padhye, Prasad Dandawate

**Affiliations:** 1Department of Cancer Biology, University of Kansas Medical Center, Kansas City, KS 66103, USA; 2MIT School of Bioengineering, Sciences & Research, MIT Art, Design and Technology University, Pune 412201, India; 3Division of Medical Oncology, University of Kansas, Kansas City, KS 66160, USA; 4Pathology and Laboratory Medicine, University of Kansas, Kansas City, KS 66160, USA; 5Institute for Advancing Medical Innovation, University of Kansas Medical Center, Kansas City, KS 66160, USA; 6Interdisciplinary Science & Technology Research Academy (ISTRA), Azam Campus, University of Pune, Pune 411001, India

**Keywords:** curcumin, difluorinated curcumin, CDF, pancreatic cancer, PDAC, 2-hydroxypropyl-β-cyclodextrin, cyclodextrin

## Abstract

Pancreatic ductal adenocarcinoma (PDAC) is the primary reason for cancer-related deaths in the US. Genetic mutations, drug resistance, the involvement of multiple signaling pathways, cancer stem cells (CSCs), and desmoplastic stroma, which hinders drug penetrance, contribute to poor chemotherapeutic efficacy. Hence, there is a need to identify novel drugs with improved delivery to improve treatment outcomes. Curcumin is one such compound that can inhibit multiple signaling pathways and CSCs. However, curcumin’s clinical applicability for treating PDAC is limited because of its poor solubility in water and metabolic instability. Hence, we developed a difluorinated curcumin (CDF) analog that accumulates selectively in the pancreas and inhibits PDAC growth in vitro and in vivo. In the present work, we developed its 2-hydroxy-propyl-β-cyclodextrin (HCD) inclusion complex to increase its water solubility and hydrolytic stability. The CDFHCD inclusion complex was characterized by spectroscopic, thermal, and microscopic techniques. The inclusion complex exhibited increased aqueous solubility, hydrolytic stability, and antiproliferative activity compared to parent CDF. Moreover, CDF and CDFHCD inhibited colony and spheroid formation, and induced cell cycle and apoptosis in PDAC cell lines. Hence, CDFHCD self-assembly is an efficient approach to increase water solubility and anticancer therapeutic efficacy, which now warrants advancement towards a clinical proof of concept in PDAC patients.

## 1. Introduction

Pancreatic ductal adenocarcinoma (PDAC) is the 4th highest cause of cancer-related deaths in the US, with a poor five-year survival 11% [1,2]. PDAC is characterized by rapid progression, invasiveness, and profound drug resistance, resulting in poor treatment outcomes [1,2]. PDAC is expected to be the second leading reason for deaths associated with cancer by the end of this decade [3]. Despite the extensive research conducted in diagnostic and therapeutic PDAC treatment, it remains a significant problem worldwide [4,5]. Moreover, the existence of germline and acquired genetic mutations such as KRAS, TP53, CDK2NA, and SMAD4/DPC4 are associated with an even poorer prognosis [6]. The existence of cancer stem cells (CSCs) [7], the occurrence of tumor-associated immune cells in the tumor microenvironment, the activation of multiple signaling pathways, and desmoplastic stroma can result in drug resistance or cancer recurrence [8,9]. The current chemotherapy for PDAC consists of gemcitabine combined with nab paclitaxel [10] or 5-fluorouracil/leucovorin with irinotecan and oxaliplatin (FOLFIRINOX) [11,12], which only improves overall survival between 8.5 and 11 months with a response rate of 23–31% [10]. Hence, it is essential to identify novel targets that are more effective treatments and that improve survival for PDAC patients.

Nature is a rich source of compounds called phytochemicals, which inhibit multiple signaling pathways, multiple cell types, and CSCs and their signaling pathways [13,14]. Phytochemicals have gained wide attention in the last 20 years because of their excellent safety profiles and ability to target several signaling pathways in cancer cells [13,14]. We have synthesized and studied analogs of phytochemicals for their anti-cancer activities, including curcumin [15], resveratrol [16,17,18], honokiol [19], plumbagin [20,21,22], cucurbitacin B and I [23], celastrol [24], triptolide [24], chalcones [25], mangostin [26], and quinomycin A [27]. We have summarized the biological activities of these phytochemicals and their analogs against various cancers in recent review articles [13,14,28,29,30,31,32,33,34,35].

We selected curcumin from the plant *Curcuma longa*. Curcumin has been found to be effective against various cancers; however, its poor bioavailability and poor water solubility have limited its clinical utility [13,14,28,29,30]. To overcome these limitations, we developed a new chemical analog of curcumin, 3,4-difluoro-benzo-curcumin, commonly known as CDF. CDF showed superior bioavailability, effective delivery to and uptake into pancreatic tissues, and the inhibition of the PDAC cell growth [36,37].

Cancer studies investigating CDF have demonstrated its effects on multiple cancer-associated pathways and phenotypes. CDF significantly inhibited the sphere-forming ability (pancospheres) of PDAC cells by down-regulating the cancer stem cell (CSC) markers EPCAM and CD44. CDF treatment inhibited tumor growth and the expression of cyclooxygenase-2 and miR-21 in a mouse xenograft model, while increasing both PTEN and miR-200. The up-regulation of miR-200 in tumors remained [38,39], and the reduction in miR-21 resulted in the induction of PTEN [40]. The CDF compound inhibited VEGF and IL-6 production and, when in hypoxic conditions, further reduced Nanog, Oct4, and EZH2 expression and miR-210 and miR-21 levels in PDAC cells [41]. CDF inhibited cell growth [42] in prostate cancer as well as chemo-resistant colon cancer cells by eradicating CSCs [43]. It inhibits the growth of 5-Fluorouracil and oxaliplatin-resistant colon cancer cells by down-regulating miR-21 levels and restoring PTEN levels with decreased p-Akt levels [44,45,46]. Recent reports have also shown that CDF inhibits MMP9 expression and activity in 549 and H1299 NSCLC cells [47]. Basak and coworkers [48] reported the anticancer activity of CDF delivered via a liposomal formulation in cisplatin-resistant head and neck squamous cell carcinoma CSCs. Recently, CDF-folic acid-conjugated polymeric micelles have also been shown to have inhibitory effects on ovarian and cervical cancer cells by inhibiting NF-kB and causing significant apoptosis [49].

Although CDF showed remarkable anticancer activity compared to curcumin in vitro, poor water solubility and metabolic instability present significant barriers to advancing this compound to clinical proof of concept. Cyclodextrin inclusion complexes have been used to enhance the water solubility, improve the bioavailability and enhance the biological activity of several phytochemicals and their synthetic analogs [15,22,50]. Yallapu and coworkers [51] have recently reported on the enhanced therapeutic activity of the β-cyclodextrin–curcumin inclusion complex compared to free curcumin against prostate cancer. Previously, pharmacokinetic studies conducted in our laboratory showed that CDF-β-cyclodextrin conjugate increases the systemic bioavailability of CDF from 6 ng/mL to 110 ng/mL, as well as tissue uptake from 300 ng/mL to 410 ng/mL in the serum and the pancreas, respectively. Moreover, we could not detect CDF following four hours of administration. In contrast, we noticed the CDF- β-cyclodextrin inclusion complex at 35 ng/mL and 280 ng/mL in serum and the pancreas, respectively. These data indicated that β-cyclodextrin conjugation increased the solubility and stability of CDF in the blood and pancreas [15]. Although β-cyclodextrin conjugates have been extensively used to improve the solubility of the drugs, toxicities are associated with clinical use [52,53]. However, the 2-hydroxypropyl derivative of β-cyclodextrin (HCD) is well tolerated when dosed orally and utilized in several approved drug products [54,55]. Hence, we prepared and assessed the anticancer properties of the CDF 2-hydroxypropyl-β-cyclodextrin (HCD) inclusion complex against PDAC cells.

## 2. Results and Discussion

### 2.1. Synthesis of CDF-2-Hydroxypropyl-Cyclodextrin (HCD) Inclusion Complex

CDF was synthesized and spectroscopically characterized per our previously reported method [37]. Based on phase solubility studies, we described previously that CDF forms 1:2 complexes with β-cyclodextrin derivatives [15]. Hence, we prepared the CDFHCD inclusion complex in a 1:2 ratio using the kneading method and used it for further spectroscopic characterization and biological assays.

### 2.2. Infra-Red Studies

Fourier Transform Infrared (FTIR) spectroscopy was used to study the interaction of guest molecule CDF and HCD within the inclusion complex [56]. The CDF spectrum (Figure 1A,B) showed absorption bands in the region 3439.19–3302.24 cm^−1^, suggestive of phenolic –OH stretching, while other bands were observed at 2596.97 (C-H, OCH_3_), 1629.90–1593.25 (C=O, C=C), 1429.30 (C-H, olefinic), 1274.99 (C-F), 1184.33–1163.11 (Ar C–O), and 1037.74–823.63 (C–O–C), respectively. HCD spectrum showed major peaks at 3410.26–3329.25 and 2968.55–2926.11 of O-H and C-H bands, whereas 1410.01 and 1371.43 cm^−1^ bands indicated the C-H stretches from CH_2_ and CH_3_. Other peaks at 1330.93, 1151.54–1082.10, 1037.74, and 947.08 cm^−1^ indicated the existence of skeletal vibrations containing α-1,4 linkages of glucose and cyclodextrin. The FTIR spectra of the CDFHCD inclusion complex showed lower or higher wavenumber shifts in the absorption frequencies of major bands for HCD and CDF (Figure 1A,B), while the decreased sharpness or disappearance of the peaks were indicative of the typical characteristics of inclusion complex formation [57]. Within the CDFHCD complex itself, the peak of the phenolic hydroxyl group appeared at 3439.19–3302.24 compared to HCD alone, which exhibited peaks at 3410.26–3329.25 cm^−1^. Similarly, the C-H and C–C–H, C–O, and C–C peaks at 2596.97 and 854.49 were shifted to 2968.55–2926.11 cm^−1^ and 844.85 cm^−1^, respectively, compared to HCD. The minor shifts in the FTIR peaks of HCD and CDF in the CDFHCD complex indicated the successful formation of the CDFHCD inclusion complex [50,58].

### 2.3. Differential Scanning Calorimetric (DSC) Studies

Thermochemical analysis techniques are successfully used to examine the physical state of the drug in polymer or inclusion complexes [59,60,61]. We used DSC to study the thermal behavior of CDF and HCD in the CDFHCD inclusion complex (Figure 1C). HCD and CDF showed a sharp endothermic peak at 81.44 °C (beginning at 33.56 and ending at 123.96 °C) and 219.46 °C (beginning at 217.76 °C and ending at 222.06 °C), respectively, because of their melting temperature. Within the CDFHCD complex, however, the prominent melting peak of CDF at 219.46 °C disappeared, and a small peak appeared at 199.78 (with onset at 198.38 °C and end at 213.62 °C). Moreover, the peak of HCD was shifted from 81.44 °C to 74.75 °C (beginning at 27.94 °C and ending at 109.21 °C). The shift in/disappearance of the endothermic peaks of CDF and HCD indicated the successful formation of the CDFHCD complex [62,63] and demonstrated stronger solid-state interactions [51].

### 2.4. Nuclear Magnetic Resonance Spectroscopy

NMR spectroscopy aids the study of the interactions between the drug and guest molecule within the cyclodextrin cavity since the electronic and chemical surroundings of the protons of both the guest and host molecules are affected during complex formation and are reflected through the shifts in the δ values of the protons [64]. In the present study, the HCD protons undergo a considerable change to downfield (higher ppm) in the CDFHCD inclusion complex, suggesting a weak interaction (Van der Waals forces and hydrogen bonding) between HCD and CDF (Figure 2A) in the inner side of the HCD cavity. The upfield shift of the protons located within the CDF and HCD cavities indicate a major hydrophobic interaction [65]. An H^1^-NMR analysis of HCD exhibited extensive resonances overlap in the 3.20 to 3.80 ppm spectral region. It therefore only allowed the identification of clusters of signals due to the different types of glycosidic or 2-hydroxypropyl protons, which made it difficult to appropriately assign the protons; meanwhile, the appearance of the methoxy protons of CDF in the CDFHCD spectra further complicated the assignments. These peaks in the methoxy groups of the CDF were shifted from 3.798 to 3.788 ppm (−0.010), and from 3.859 to 3.849 ppm (−0.010) after the formation of the inclusion complex, respectively, suggesting the involvement of aromatic rings in the complexation of CDF with HCD. The aromatic ring protons of CDF appeared at 6.77–7.64 ppm, which showed an upfield shift in the region of 6.75–7.63 ppm in the CDFHCD spectra, suggesting their involvement in the HCD complex formation. Nonetheless, the peaks of CDF between 6 and 8 ppm in the CDFHCD spectra were not clear enough to assign the shifts; this was due to the merging and reduced multiplicity of the peaks, which is characteristic of an inclusion complex [51,66,67]. These data indicated the successful formation of a complex between CDF and HCD, while it can be proposed that both aromatic rings of CDF participated in the inclusion of the HCD cavity.

### 2.5. Scanning Electron Microscopic (SEM) Studies

SEM was utilized to analyze the bulk surface morphology of the CDF, HCD, and CDFHCD inclusion complexes (Figure 2B). The analysis indicated that HCD exhibited amorphous, ‘shrunken’ spheres and flakes, while CDF showed rod-like-shaped or hexagonal crystalline structures. The CDFHCD inclusion complex did not show flakes or rod-shaped crystals but presented irregularly shaped aggregates or clump formations. This modification in the morphology of the CDFHCD inclusion complex has been ascribed mainly to the inclusion of CDF in the HCD cavity. These data suggest the formation of the CDFHCD inclusion complex [68].

### 2.6. Molecular Docking Studies

We used the molecular docking technique to further study the CDF’s binding mode with the HCD cavity in the inclusion complex [15,58,69]. We found that CDF can interact with HCD in a 1:2 ratio. The possible stable structure of the CDFHCD (1:2) interaction is shown in Figure 2C. CDF was predicted to bind with HCD (1:2) with the binding energy of −6.3 Kcal/mol, while the methoxy and hydroxyl groups on the aromatic ring were found to interact with the HCD cavity via two hydrogen bonds (2.4 Å) (Figure 2C). These observations agree with the NMR study predictions described in previous reports [15].

### 2.7. Anticancer Activity

#### 2.7.1. CDF and CDFHCD Inhibit the Proliferation of PDAC Cells

First, we studied the effects of CDF and CDFHCD (dose range 0–5 µM) on the proliferation of PDAC cell lines based on their mutation, origin, and metastatic potential. Both CDF and CDFHCD inhibited (time- and dose-dependent effects) the proliferation of MiaPaCa-2 (KRAS^G12C^, p53^R248W^), Panc-1 (KRAS^G12D^, p53^R273H^), Panc 01728 (patient-derived cell line), and S2-007 (Metastatic) cells (Figure 3A). The IC_50_ values of CDF and CDFHCD against PDAC cell lines at different time points are summarized (Figure 3B). We observed the significant increase in the antiproliferative activity of CDF after HCD inclusion complexation (Figure 3B, *p* < 0.05), which can be attributed to a higher water solubility and a higher uptake of CDF into pancreatic cancer cells through the formation of the cyclodextrin inclusion complex [15]. We then used the clonogenic assay to study the long-lasting effects of CDF and CDFHCD on PDAC cell lines. MiaPaCa-2 and Panc-1 cells were treated with CDF and CDFHCD at IC_50_ and ½IC_50_ concentrations for 24 h and 48 h. After 48 h, the cell culture media containing CDF or CDFHCD was replaced with fresh drug-free media and grown for 10–14 days. Both CDF and CDFHCD treatment significantly (*p* < 0.01) reduced the colony formation (both size and number) in MiaPaCa-2 and Panc-1 cells (*p* < 0.01) (Figure 4A–D), suggesting that the anticancer effects of CDF and CDFHCD are non-reversible. Further, we evaluated the effect of CDF and CDFHCD on HPNE cells (immortalized epithelial pancreatic ductal cell line) to characterize the cytotoxicity potential. While CDF produced cytotoxicity in HPNE cells at a 1 μM concentration, CDFHCD did not induce any toxicity up to a 5 μM concentration (Appendix A). This suggests the potentially lower cytotoxic effects on immortalized cells. However, more detailed studies with different normal cell lines are needed in the future to evaluate this. Given the greater efficacy of the CDFHCD inclusion complex, we used IC_50_ concentrations of CDF and CDFHCD for further studies in order to understand whether there are differences in their mechanism(s) regarding anticancer activity.

#### 2.7.2. CDF and CDFHCD Induce Cell Cycle Arrest

We studied the effects of CDF and CDFHCD on cell cycle progression using flow cytometry. The treatment of CDF and its inclusion of complex CDFHCD caused a significant (*p* < 0.01) increase in the number of cells in the sub-G0 phase at 24 h and 48 h in PDAC cell lines (Figure 5A,B and Appendix A). Moreover, the concentration of CDF that is equivalent to CDFHCD produced less potent effects than CDFHCD. The overexpression of Cyclin D1 plays a role in cancer progression and development [70] by controlling the cell cycle progression. Hence, we performed the western blot analysis and observed the significant (*p* < 0.01) downregulation of cyclin D1 after CDF and CDFHCD treatment (Figure 5C,D). These data suggested that CDF and CDFHCD induce cell cycle arrest in PDAC cells.

#### 2.7.3. CDF and CDFHCD Induce Apoptosis

The accumulation of cancer cells in a sub-G0 stage after CDF and CDFHCD treatment can result from DNA fragmentation, suggesting probable cytotoxic effects. To study this, we utilized the Annexin V/PI assay with flow cytometry. We observed an increased cell number over 48 h in the late apoptosis stage after CDF and CDFHCD treatment compared to the control cells (Figure 6A and Appendix A). We observed CDF autofluorescence interfering in early apoptosis readings differentially; hence, we cannot decisively determine the role that CDF and CDFHCD play in early apoptosis. We further confirmed apoptosis using caspase 3/7 assays to examine the effector activity of caspase. CDF and CDFHCD treatment increased the caspase 3/7 activity (Figure 6B, *p* < 0.01) in PDAC cells. Mechanistically, the western blot analysis showed the enhanced expression of cleaved PARP protein (*p* < 0.05) in CDF and CDFHCD-treated cells (Figure 6C and Appendix A). The CDF and CDFHCD treatment reduced the levels of the anti-apoptotic markers Mcl1 (*p* < 0.05) and Bcl2 (*p* < 0.01), while proapoptotic protein Bax expression was not significantly changed (Figure 6C). A decrease in Bcl-2 levels is established as a mechanism of apoptosis in cancer cells [71]. These data suggest the utility of CDF and CDFHCD in combination with the first-line chemotherapeutic drugs [72,73]. These data indicate that the CDF and its inclusion in the CDFHCD complex can induce apoptosis in PDAC cells.

#### 2.7.4. CDF and CDFHCD Inhibit the Spheroid Formation

Increasing evidence suggests that CSCs are involved in causing drug resistance, aggressiveness, and recurrence in PDAC [74]. Several surface markers that mark CSCs, including CD44, have been identified in PDAC [75]. Moreover, the overexpression of CD44 induces the activation of cMyc, which in turn activates the MEK and ERK pathway to inhibit apoptosis in cancer cells [76]. cMyc is also overexpressed in CSCs in order to maintain the pluripotency [77], and its overexpression is involved in PDAC progression and drug resistance [78]. Recent studies have also shown that doublecortin-like kinase 1 (DCLK1) is a novel PDAC CSC marker [79]. Hence, to completely eradicate cancer, it is essential to understand CSCs’ growth and develop novel therapeutic agents that target those mechanisms. It is reported that CSCs form spheroids in ultra-low attachment plates in suspension culture. Hence, we used a spheroid formation assay to analyze the effect of CDF and CDFHCD on CSCs. CDF and CDFHCD treatment reduced the spheroid formation (size and number) in MiaPaCa-2 and Panc-1 cells (Figure 7A,B). Moreover, CDF and CDFHCD treatment downregulated the expression of CD44 (*p* < 0.01), DCLK1 (*p* < 0.01) and cMYC (*p* < 0.01) in MiaPaCa-2 and Panc-1 cells (Figure 7C and Appendix A). These datasets showed that CDF and CDFHCD inhibit spheroid growth and CSC marker protein expression.

### 2.8. Hydrolytic Stability Study

Curcumin is prone to hydrolysis and is hydrolyzed to ferulic acid, feruloyl methane, and vanillin [80]. Cyclodextrin drug inclusion complexes, however, have been reported to improve the hydrolytic stability of guest drug molecules [81]. Hence, we performed the hydrolytic stability study of CDF with and without HCD in PBS solution [82] to study the hydrolysis of CDF. The UV-visible spectra of CDF showed hydrolysis from 15 min to 1 h in PBS, as demonstrated by a reduction in the intensity of the absorption bands. In comparison, we observed complete hydrolysis within 1 h (Figure 8A). CDF showed resistance to hydrolysis up to 5 h in PBS with 10% HCD (Figure 8B). The binding constant (K_D_) of CDF in 10% HCD solution was calculated to be 4.50 ± 0.27 × 10^−10^ M^−1^, which suggests the excellent stability of the complex for potential in vivo application. The study indicated that HCD complexation improved the hydrolytic stability of CDF.

## 3. Materials and Methods

### 3.1. Synthesis of CDF

The difluorinated curcumin analog CDF was synthesized and characterized using our previously published method [37]. The curcumin was separated from the mixture of curcuminoids by column chromatography using dichloromethane and methanol in a 9:1 proportion (*v*/*v*). The first fraction was found to be curcumin and was used for further reactions. The purified curcumin (1 mmol) was dissolved in methanol, and 3,4-difluoro-benzaldehyde (1 mmol) was added dropwise to this reaction with constant stirring in the presence of the catalytic amount of piperidine. The reaction was further stirred for 48 h and was monitored by TLC. The purified fraction by column chromatography was dried under a vacuum.

### 3.2. Preparation of Inclusion Complexes

We have previously shown that CDF forms a 1:2 complex with β-cyclodextrin by using phase solubility studies. We used the kneading method for making CDFHCD inclusion complexes. CDF and HCD were mixed in the proportion of 1:2 molar concentrations in a mortar for one hour by adding a mixture of methanol and deionized water (1:1) to get a slurry-like uniformity using a pestle. The slurry was further dried in an oven. The dried complex was used for further studies [15].

### 3.3. Infra-Red Studies

CDF, HCD, and CDFHCD were subjected to infra-red studies using the potassium bromide disk method and using the Shimadzu Fourier Transform Infrared (FTIR)-8700 spectrophotometer. A transparent disk of CDF, HCD and CDFHCD was made by mixing potassium bromide using high pressure that was applied by dyes. The resultant disk was positioned in an IR spectrophotometer, and the spectrum was recorded from 4000 to 400 cm^−1^.

### 3.4. Differential Scanning Calorimetric (DSC)

CDF, HCD, and CDFHCD were studied using a DSC (Mettler Toledo, Switzerland). Then, 2.5–5 ± 0.5 mg of sample was positioned in sealed aluminum pans in the presence of liquid nitrogen as a cooling agent. The thermograms were recorded by scanning from 20 to 300 °C at ten °C/min intervals.

### 3.5. Nuclear Magnetic Resonance Spectroscopy

The proton NMR spectra were recorded in DMSO-d6 by the BRUKER AV III (^1^H-NMR 500 MHz) spectrophotometer. The chemical shifts (δ) were expressed in parts per million (ppm) using residual solvent DMSO as a reference (2.50 ppm). The spectra were generated using Bruker’s TOPSPIN-2.1 software. NMR experiments were conducted by setting the scan numbers at 64, the relaxation delay at 1.0 s, and the pulse degree at 25 °C.

### 3.6. Molecular Docking Studies

The 3D structure of CDF was prepared using CORINA [83] software from smiles that were further energy minimized in the PRODRG server [84]. The β-CD was extracted from the 3D structure of alpha-amylase (PDB id: 1JL8.pdb) and converted to HCD by adding 2-hydroxypropyl groups. The HCD was then prepared for docking by adding polar hydrogen and energy minimized. For studying CDFHCD interaction in a 1:2 ratio, the two HCD molecules were prepared and energy minimized in VMD software 1.9.3 [85] and the PRODRG server, respectively. Molecular docking was carried out to obtain the possible binding modes for the CD-HCD complex using AutoDock Vina software version 1.2.3 [86]. The grid was created around the *X*, *Y*, and *Z*-axis (40 × 40 × 44) using the Lamarckian Genetic Algorithm (LGA). The stable CDFHCD conformation was selected by considering the lowermost binding energy and hydrogen bonds formed between CDF and HCD. The CDFHCD complex was analyzed and visualized using Pymol software version 1.7.4 [87].

### 3.7. Cell Culture

The PDAC cells MiaPaCa-2, Panc-1 (obtained from American Type Culture Collection), Panc 01728 (a patient-derived cell line, a gift from Dr. Shrikant Anant’s lab), HPNE and S2-007 (a gift from Dr. Animesh Dhar laboratory) were cultured in Dulbecco’s Modified Eagle Medium (DMEM), supplemented with L-glutamine, 4.5 g/L of glucose, and sodium pyruvate (Corning, Tewksbury, MA, USA). Finally, heat-inactivated fetal bovine serum (10% concentration, FBS) (Sigma-Aldrich, St. Louis, MO, USA) and 1% antibiotic-antimycotic solution (Corning, Tewksbury, MA, USA) were added to make complete media. PDAC cells were grown in a 5% CO_2_-humidified at 37 °C. PDAC cell lines were within 20 passages and validated by STR analysis. All methods followed the standard guidelines, regulations and the manufacturer’s instructions.

### 3.8. Proliferation and Colony Formation Assays

For the proliferation and colony formation assays, 5 × 10^3^ cells/well PDAC cells (MiaPaCa-2, Panc-1, S2-007, P01728) were plated in 96-well plates and grown in complete media. Cells were treated with increasing doses of CDF, CDFHCD, and the respective controls (DMSO and HCD). A hexosaminidase assay field recorded the cell viability at different times (24 h, 48 h, and 72 h) [88]. The percent of growth inhibition was calculated by comparing cell viability with the controls, while IC_50_ values were calculated by plotting a graph of % viability vs. concentration. The concentration that showed 50% cell viability was considered an IC_50_ value. In total, 500 cells/well of PDAC cells were plated in complete DMEM media for the colony formation assay. PDAC cells were treated with different doses of CDF and CDF–HCD. Media was changed following 24 h and 48 h of CDF and CDFHCD treatment to remove drug exposure. Further, these cells were grown and they formed colonies for 10–12 days. The resulting colonies were washed and fixed using formalin (10% solution). Further, formalin was aspirated, washed with PBS, and stained using a staining solution (1% crystal violet solution in 10% ethanol). The plates containing colonies were dried at room temperature, counted, and compared to untreated cells [23].

### 3.9. Cell Cycle Analysis by Flow Cytometry

For the cell cycle analysis, 5 × 10^5^ PDAC cells/well (MiaPaCa-2 and Panc-1) were plated in a 10 cm cell culture dish and treated with CDF and CDFHCD at indicated time points. Cells were trypsinized, washed, resuspended in PBS, fixed using 70% ethanol in PBS, and stored at 4 °C overnight after the indicated time points. These cells were further permeabilized and stained with FxCycle^TM^ PI/RNase staining solution (Invitrogen, Eugene, OR, USA) at room temperature and subjected to flow cytometry. Flow cytometric analysis was performed with a FACS Calibur analyzer (Becton Dickinson, Mountain, View, CA, USA) and studied using BD FACSDiva 8.0.1 software (Verity Software House, Topsham, ME, USA) 

### 3.10. Apoptosis Assays

For the first assay, the Apo-one Homogeneous Caspase-3/7 Assay kit (Promega Corporation, Madison, WI, USA) was used following the manufacturer’s instructions in order to estimate caspase 3/7 activity in PDAC cells (MiaPaCa-2 and Panc-1) after CDF and CDFHCD treatment. In the second assay, Annexin V/PI staining was performed and studied using flow cytometry. Briefly, 1 × 10^5^ PDAC cells were plated in 6-well plates. After 24 h of plating, PDAC cells (MiaPaCa-2 and Panc-1) were treated with vehicle control, CDF, and CDFHCD at IC_50_ doses. At the end of the treatment, cells were washed and stained with Annexin V-FITC antibody and propidium iodide (PI), as per the manufacturer’s protocol, and were studied by flow cytometry.

### 3.11. Spheroid Formation Assay

In total, 500 PCAC cells were plated in each well of the ultralow attachment plates (Corning, Lowell, MA, USA). The spheroid media was composed of DMEM medium added to FGF (20 ng/mL), EGF (20 ng/mL), heparin (4 ug/mL), pen/strep (1%) (Invitrogen), and B27 (10 mL in 500 mL of 50×), which was used for growing spheroids. Spheroids were treated with vehicle, CDF, and CDFHCD at IC50 and ½IC50 concentrations. Spheroids were counted and imaged after five days.

### 3.12. Western Blot Analysis

For the western blot analysis, 5 × 10^5^ cells PDAC cells were plated in 10 cm dishes. After 24 h of plating, cells were treated with vehicle, CDF, and CDFHCD for 48 h at IC50 concentrations. Cell lysis was then performed in a protein lysis buffer and protease inhibitor (ThermoScientific, Rockford, IL, USA) solution and sonicated to achieve complete lysis. The resultant lysates were centrifugated at 6000 rpm for 10 mins at a cold temperature. The protein estimation was performed using a BCA reagent (Pierce™ BCA Protein Assay Kit) (ThermoScientific, Rockford, IL, USA), and 50 µg of protein for each group was used to load into the gels. Protein lysates were separated using polyacrylamide gel electrophoresis and transferred onto Immobilion membranes (PVDF, Millipore, Bedford, MA, USA). Post transfer, the membranes were removed from the transfer assembly, blocked using 5% milk for 1 h, washed, and probed with primary antibodies. The individual proteins were identified using the chemiluminescence system (GE Health Care, Piscataway, NJ, USA). The Bio-Rad ChemiDoc-XRS+ instrument quantified the data using image lab software and recorded the protein levels. All antibodies (Cyclin D1 (CST#2922), Bax (CST#2772), Bcl2 (CST#4223), Bcl-XL (CST#2762), Mcl-1 (CST#4572), PARP (CST#9542), CD44 (CST#3570), and cMyc (CST#9402) were purchased from Cell Signaling Technology (Beverly, MA, USA), and GAPDH (G-9) was purchased from Santa Cruz Biotech, Inc. (Santa Cruz, CA, USA). The DCLK1 antibody was purchased from Sigma-Aldrich (Sigma#SAB4200186). The antibodies were diluted per the manufacturer’s instructions (1:1000 dilution in 5% BSA in TBST).

### 3.13. Hydrolytic Stability Study

The hydrolytic stability study used the previously reported method [50]. The stock solution of CDF was made in DMSO. Then, 100 μL of the stock solution was added to 10 mL of the 10% (*w*/*v*) HCD in PBS and PBS alone, respectively. The absorption spectra were recorded by taking aliquots at indicated intervals up to 72 h.

### 3.14. Statistical Analysis

All experiments were repeated three times, and the experiment values are shown as the mean  ±  SD. The experimental datasets were examined using an unpaired two-tailed *t*-test comparing it to the control group. The western blot quantifications were analyzed using a one-way ANOVA test, which was compared to the control group using GraphPad’s Prism-9 (Boston, MA, USA). A *p* < 0.05 was chosen to be statistically significant.

## 4. Conclusions

In the present study, we successfully prepared and characterized the inclusion complex of CDF with 2-hydroxypropyl-β-cyclodextrin, CDF–HCD. CDFHCD treatment demonstrated greater anti-proliferative effects against PDAC cell lines compared to CDF treatment alone. CDF and CDFHCD inhibited colony and spheroid formation, producing cell cycle arrest and apoptosis in PDAC cell lines. Moreover, HCD improved the hydrolytic stability of CDF. Future studies will be directed toward characterizing the ability of the CDFHCD inclusion complex to deliver sufficient CDF into PDAC tumors, the engagement of CDF with the target(s) within PDAC cells, the effects on tumor development and progression, as well as survival in validated mouse models of PDAC.

## Figures and Tables

**Figure 1 ijms-24-06336-f001:**
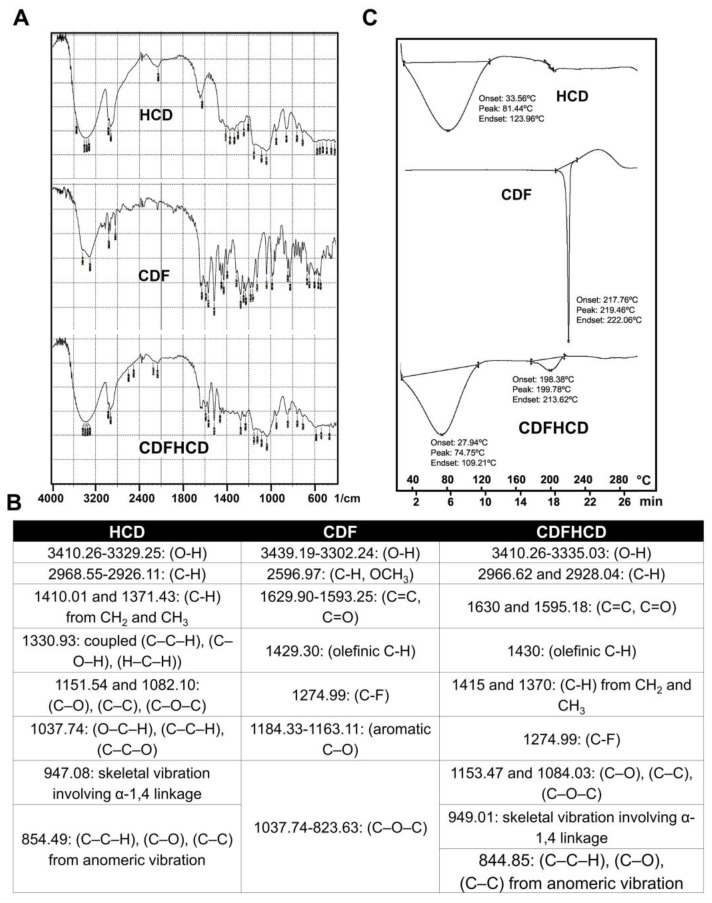
Spectroscopic and thermal Characterization of CDFHCD inclusion complex. (**A**). FTIR spectra of 2-hydroxypropyl-β-cyclodextrin (HCD), CDF, and CDFHCD inclusion complex. (**B**). The FTIR spectral assignments are summarized. (**C**). Differential scanning calorimetry (DSC) spectra of HCD, CDF, and CDFHCD inclusion complex.

**Figure 2 ijms-24-06336-f002:**
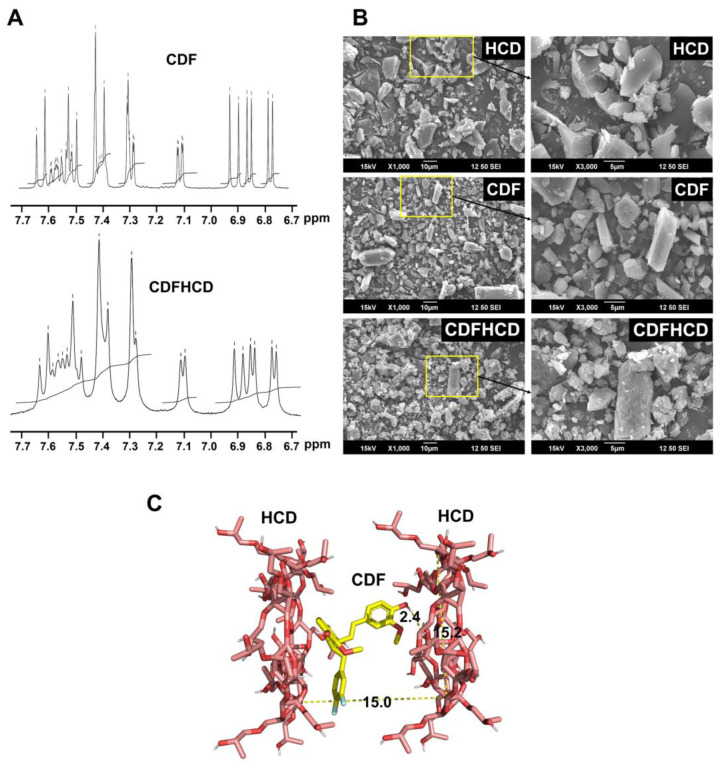
Morphological characterization of CDFHCD complex and the interactions of the CDF within the cyclodextrin cavity. (**A**). NMR spectra of 2- HCD, CDF, and CDFHCD inclusion complex. (**B**). SEM images of 2- HCD, CDF, and CDFHCD inclusion complex. (**C**). The binding mode of CDF in the cavity of HCD was proposed using molecular docking.

**Figure 3 ijms-24-06336-f003:**
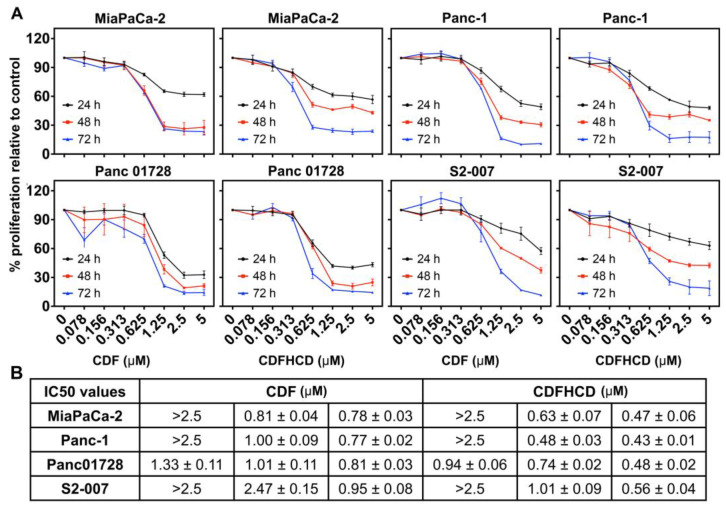
CDF and CDFHCD inhibit the proliferation of PDAC cell lines. (**A**). PDAC cells (MiaPaCa-2, Panc-1, Panc01728, and S2-007) were treated with increasing concentrations of CDF and CDFHCD (0–5 μM) for up to 72 h and studied using hexosaminidase assay. (**B**). The IC_50_ values of the antiproliferative activity are summarized in the tabular format as mean ± SD.

**Figure 4 ijms-24-06336-f004:**
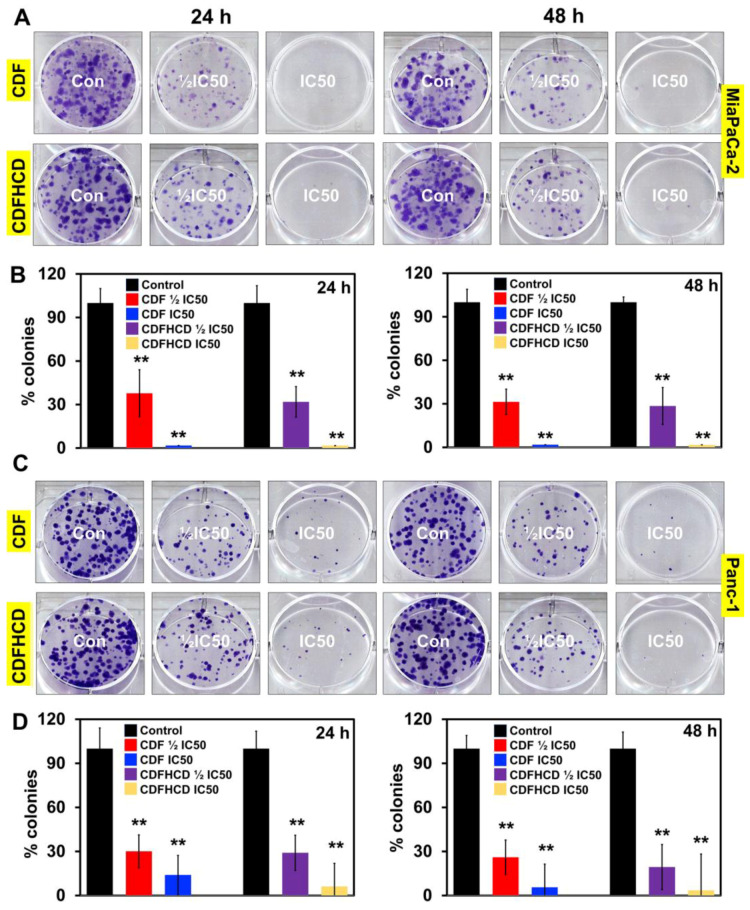
CDF and CDFHCD inhibit the colony formation of PDAC cell lines. (**A**). MiaPaCa-2 and (**C**). Panc-1 cells were incubated with ½ IC50 and IC50 concentrations of CDF and CDFHCD for 48 h and grown into colonies. Treatment with CDF and CDFHCD inhibited the number of colonies in (**B**) MiaPaCa-2 and (**D**) Panc-1 cells (** *p*  <  0.01).

**Figure 5 ijms-24-06336-f005:**
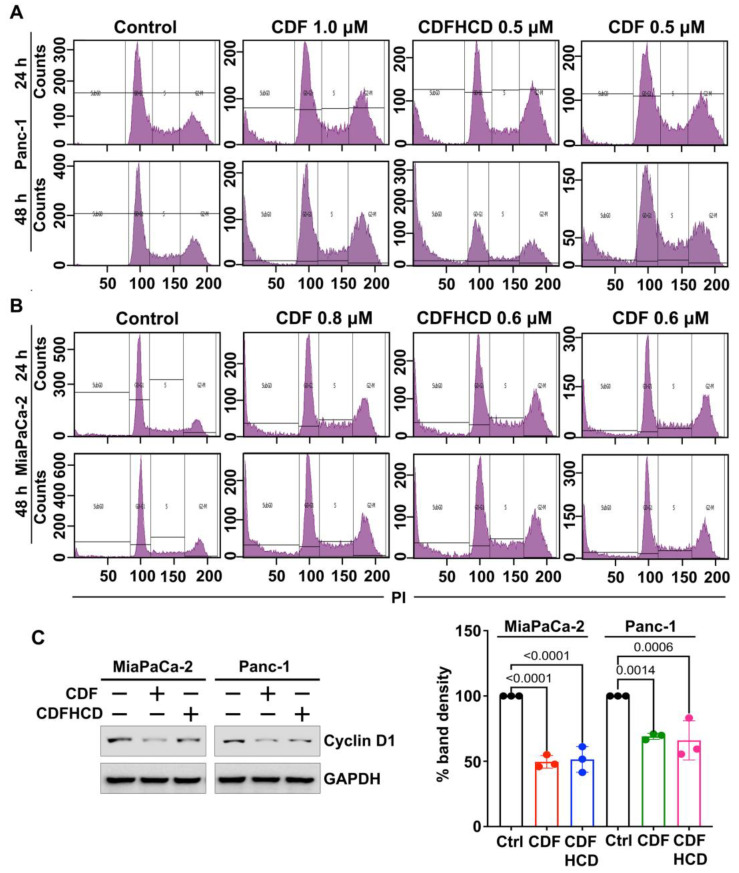
Cell cycle analysis of CDF and CDFHCD in PDAC cells. (**A**) MiaPaCa-2 and (**B**) Panc-1 cells were incubated with CDF and CDFHCD for 24 h and 48 h and analyzed by flow cytometry using FxCycle^TM^ PI/RNase staining solution. (**C**) Cell lysates from CDF and CDFHCD-treated MiaPaCa-2 and Panc-1 were analyzed using western blot to study the changes in cyclin D1 expression.

**Figure 6 ijms-24-06336-f006:**
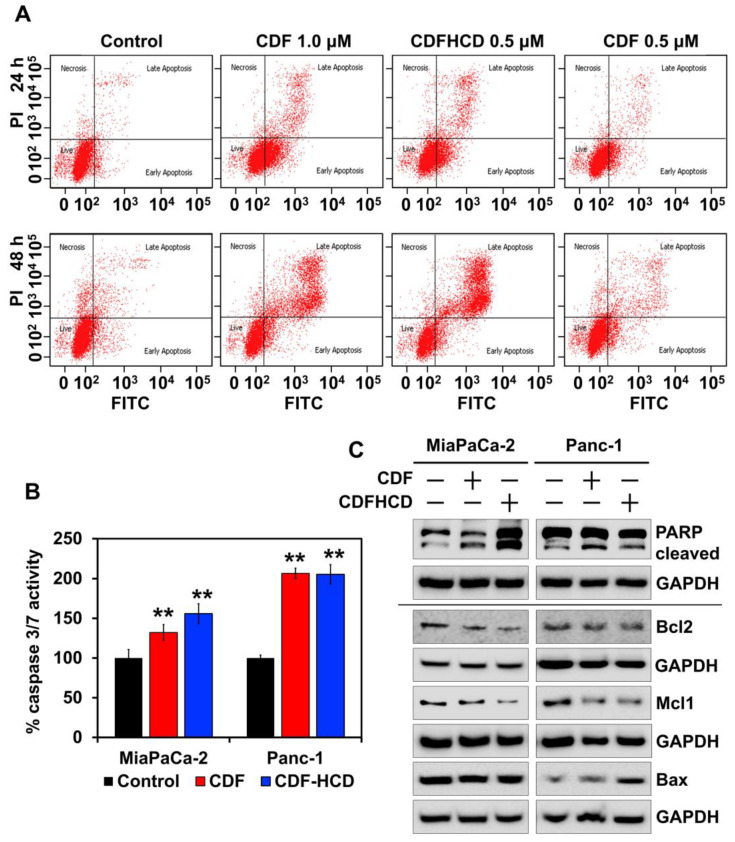
CDF and CDFHCD induce apoptosis. (**A**). Panc-1 cells were incubated with CDF and CDFHCD, stained with Annexin V-FITC and PI, and examined by flow cytometry. (**B**). Caspase 3/7 assay showed increased caspase activity in MiaPaCa-2 and Panc-1 cells at 48 h after treatment with CDF and CDFHCD (** *p*  <  0.01). (**C**). PDAC cell lysates treated with CDF and CDFHCD were analyzed by western blot to study the changes in the expression of apoptotic marker proteins, PARP, Mcl1, Bax, and Bcl2.

**Figure 7 ijms-24-06336-f007:**
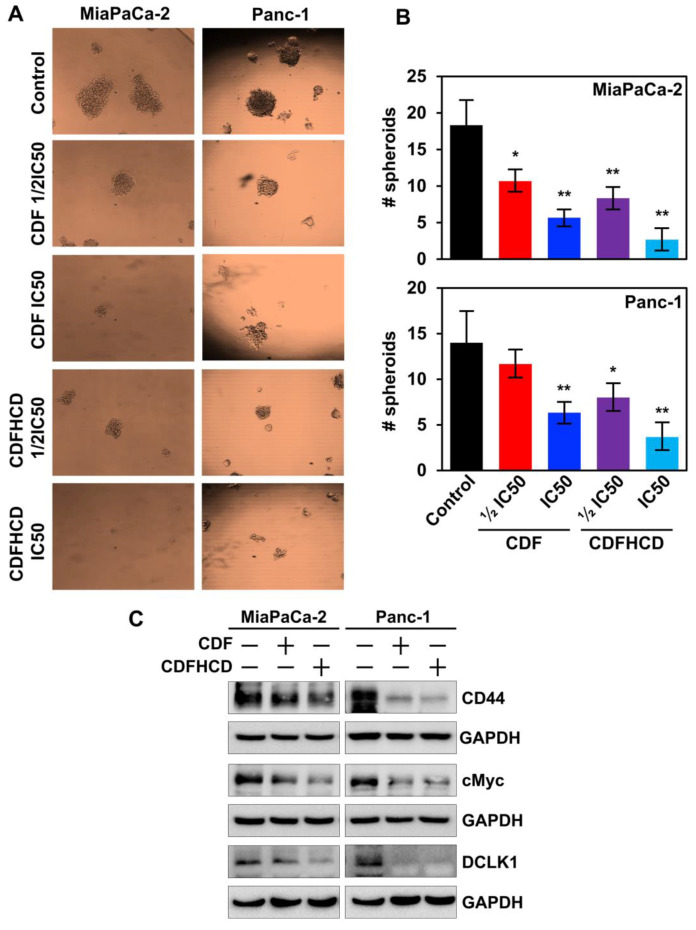
CDF and CDFHCD inhibit spheroid formation. (**A**). PDAC cells were grown in spheroid media in ultra-low attachment plates and treated with ½IC50 and IC_50_ concentrations of CDF and CDF-HCD. After five days, the pancospheres were counted and imaged (10× magnification). (**B**). CDF and CDFHCD inhibited the number of pancospheres (* *p*  <  0.05, ** *p*  <  0.01). (**C**). Cell lysates from MiaPaCa-2 and Panc-1 cells treated with CDF and CDFHCD were examined by western blot to study the changes in CEC marker proteins, CD44, cMyc, and DCLK1 expression.

**Figure 8 ijms-24-06336-f008:**
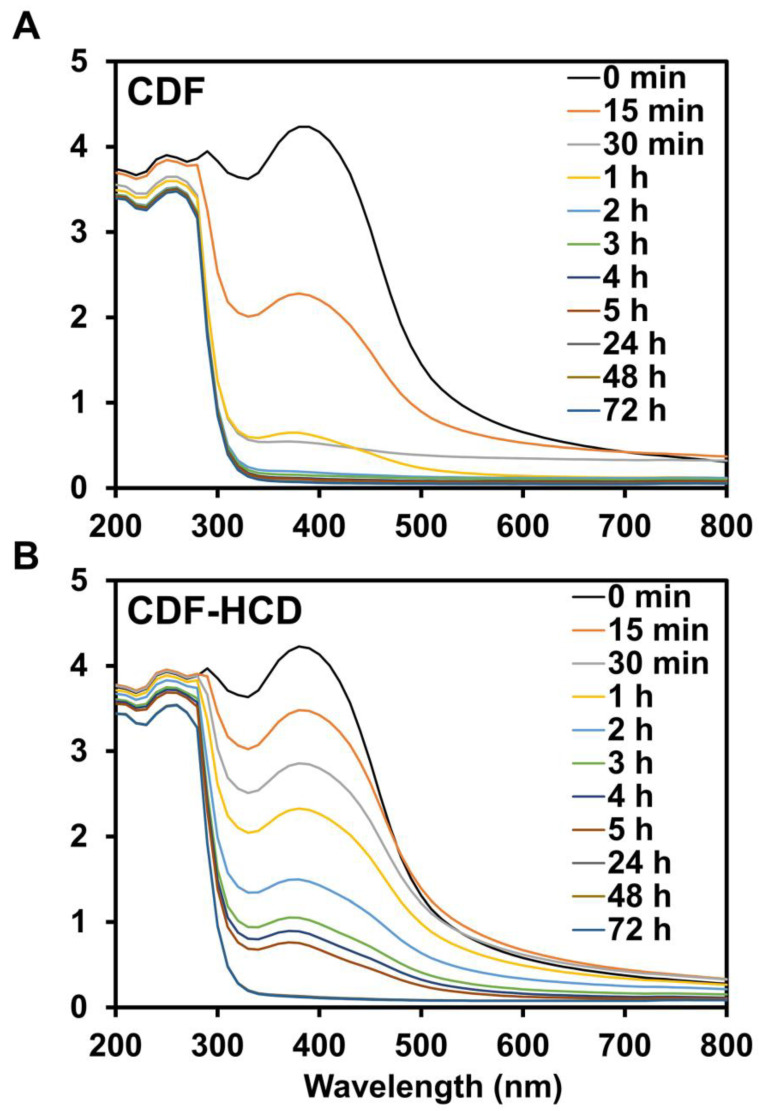
Hydrolytic stability study of CDF in PBS without (**A**) and with (**B**) 10% HCD solution over 0–72 h.

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
