# Peer review of "Anticancer Activity of Novel Difluorinated Curcumin Analog and Its Inclusion Complex with 2-Hydroxypropyl-β-Cyclodextrin against Pancreatic Cancer"

_ijms, 2023, doi:10.3390/ijms24076336_

Round 1

Reviewer 1 Report

In this submitted manuscript no# ijms-2156387, the authors, Bhattacharyya et al, tried to investigate the Anticancer Activity of Novel Difluorinated Curcumin analog and its Inclusion Complex with 2-Hydroxypropyl-β-Cyclodextrin against Pancreatic Cancer in-vitro. The manuscript is well written, and the experimental designs support their hypothesis. My observations are as follows- 

  1. In line #5, The designation of prof. Subhash B. Padhye (marked as 'f') has not been stated.   2. Fourier Transform Infrared (FTIR) spectroscopy was used to study the interaction of  CDF with HCD in the inclusion complex.  FTIR peaks of HCD and CDF after complex formation indicated the successful formation of the CDF-HCD inclusion complex.   3. In the Differential Scanning Calorimetric (DSC) Studies, the shift/disappearance in the endothermic peaks of CDF and HCD indicated the successful formation of the CDF-HCD complex and demonstrated stronger solid-state interactions.   4. The NMR data indicated the successful complex formation between CDF and HCD, while it can be proposed that both aromatic rings of CDF participated in the inclusion of the HCD cavity.   5. The SEM data demonstrated the formation of the CDF-HCD inclusion complex.    6. Molecular Docking Studies indicated possible stable interaction between CDF-HCD molecules.    7. while testing the anti-proliferative property and colony formation assay of CDF-HCD, the authors used four pancreatic cell lines, i.e. MiaPaCa-2, Panc-1, Panc-01728, and S2-007 cell lines. The authors may highlight the rationale behind selecting these cell lines.    8. The authors have successfully demonstrated that CDF and CDF-HCD can induce cell-cycle arrest and apoptosis. A westen blot of Beclin 1 will complete the panel.     9. In Figure 7-A, the scale bars are missing from spheroid formation photographs. A few more stemness markers in the WB panel will be helpful.    10. As this paper does not have a designated 'discussion', most of the data have been discussed under the result section. Little elaboration of the section -2.7.2, 2.7.3,2.7.4, and 2.8.    I have "attached" a word file containing the aforementioned comments, for your reference.   

Author Response

Reviewer 1

In this submitted manuscript no# ijms-2156387, the authors, Bhattacharyya et al, tried to investigate the Anticancer Activity of Novel Difluorinated Curcumin analog and its Inclusion Complex with 2-Hydroxypropyl-β-Cyclodextrin against Pancreatic Cancer in-vitro. The manuscript is well written, and the experimental designs support their hypothesis. My observations are as follows- 2. Fourier Transform Infrared (FTIR) spectroscopy was used to study the interaction of  CDF with HCD in the inclusion complex.  FTIR peaks of HCD and CDF after complex formation indicated the successful formation of the CDF-HCD inclusion complex.   3. In the Differential Scanning Calorimetric (DSC) Studies, the shift/disappearance in the endothermic peaks of CDF and HCD indicated the successful formation of the CDF-HCD complex and demonstrated stronger solid-state interactions.   4. The NMR data indicated the successful complex formation between CDF and HCD, while it can be proposed that both aromatic rings of CDF participated in the inclusion of the HCD cavity.   5. The SEM data demonstrated the formation of the CDF-HCD inclusion complex.    6. Molecular Docking Studies indicated possible stable interaction between CDF-HCD molecules.    

Response: We thank the reviewer for reviewing our manuscript.

Comment 1. In line #5, The designation of prof. Subhash B. Padhye (marked as 'f') has not been stated. 

Response: We corrected the affiliation of Dr. Padhye in the revised manuscript.

Comment 2. while testing the anti-proliferative property and colony formation assay of CDF-HCD, the authors used four pancreatic cell lines, i.e. MiaPaCa-2, Panc-1, Panc-01728, and S2-007 cell lines. The authors may highlight the rationale behind selecting these cell lines. 

Response: This is an excellent point. The revised manuscript has added the rationale behind selecting these PDAC cell lines. Section 2.7.1

Comment 3. The authors have successfully demonstrated that CDF and CDF-HCD can induce cell-cycle arrest and apoptosis. A westen blot of Beclin 1 will complete the panel. 

Response: Thank you for the outstanding suggestion. Beclin-1 is primarily involved in the autophagy induction in cells in response to stress. Moreover, Beclin 1-bcl2/Bcl-XL is responsible for autophagy inhibition. The current manuscript shows that CDF and CDFHCD induced apoptosis (annexin-PI and caspase 3/7 assay). Adding beclin 1 western blot to the apoptosis panel without studying other autophagy markers such as LC3B, p62, ATG genes, and PI3K family of proteins, including Vsp 15, 30, 34, will not produce the complete story. Hence, we will follow your suggestion for our future studies.  

Comment 4. In Figure 7-A, the scale bars are missing from spheroid formation photographs. A few more stemness markers in the WB panel will be helpful.  

Response: All photographs were taken at 10X magnification. Per your suggestion, we also performed a western blot with the additional stem cell marker DCLK1. We found that CDF and CDFHCD inhibit the expression of DCLK1. Please refer to section 2.7.4 and Figure 7C, Supplementary Fig 4 (quantification).

Comment 5. As this paper does not have a designated 'discussion,' most of the data have been discussed under the result section. Little elaboration of the section -2.7.2, 2.7.3,2.7.4, and 2.8.    I have "attached" a word file containing the aforementioned comments, for your reference.   

Response: Thanks. We combined the results and discussion part to ease discussing the chemistry part of the paper and followed the same for the biological studies. According to journal guidelines, authors can decide on the outline of the manuscript.

Reviewer 2 Report

In “Anticancer Activity of Novel Difluorinated Curcumin analog and its Inclusion Complex with 2-Hydroxypropyl-β-Cyclodextrin against Pancreatic Cancer”, Bhattacharyya et al, developed and characterized the inclusion complex of CDF with 2-hydroxypropyl-β-cyclodextrin which exerted greater antiproliferative effects against PDAC cell lines compared to CDF. They demonstrated that, both CDF and CDF-HCD, inhibited colony and spheroid formation, induced cell cycle arrest and apoptosis in PDAC cell lines. Interestingly, HCD improved hydrolytic stability of CDF.

Major comments:

1.       IC50 determination for proliferation was performed using 0 to 2.5 uM doses according to the figures. First, the legend figure say that cells were treated with 0 to 3 uM. Second, the table in Figure 3B show an IC50  of 2.5uM, but for accurate determination a curve with higher doses should be performed. Third, the authors should include error bars in the graphics. Also, for some cell lines, an IC50 higher to 5uM is indicated, this is not correct since the curve does not include this value. Also, the authors should analyse if the are significative differences in the IC50 of CDF-HCD compared to CDF. Finally, Materials and Methods should include how was the IC50 determined and IC50 in the table should include the mean and standard deviation.

2.       When analysed CDF—HCD effect on cell cycle arrest the authors explain that treatment of CDF and CDF-HCD caused G2/M cell cycle arrest at 24 hours, while at 48hs an increase in cell number in the sub-G0 phase was observed. However, the histograms in Figure 5 shows a marked increase in sub-G0 cells after treatments at both times, 24 and 48hs. Also, in addition with the histograms of one representative experiment, authors should include a graphbar showing the mean and standard deviation of the biological replicates and also the statistical analyses and significance.

3.       Also, in Figure 5C authors should include the quantitation of the western blot, and it would be even better to include a bar graph showing mean and stardard deviation of the biological replicates.

4.       When analyze apoptosis, authors should include, in addition with the representatives dot-plots, a bar graph depicting mean, standard deviation and statistical analyses. Also, authors claims that “We saw an increased cell number over a period of 48 h in the early and late apoptosis stage after CDF or CDF-254 HCD treatment, when compared to control cells (Figure. 6A).”. They don’t describe the results at 24hs, that seem to be similar. Evenmore, how is autofluorescence in treated cells, it seems like curcumine interfere with annexin-FITC, since a shift to the right is observed in double negative cell population. Hence, the effect in late apoptosis is clear, but I’m not sure about the percentage indicated to early apoptotic cells.

5.       In Figure 6C, density quantitation of each band should be showed and it would be better to include a bar graph showing mean, standard deviation and statistical analyzes of biological replicates.

6.       In Figure 7C, density quantitation of each band should be showed and it would be better to include a bar graph showing mean, standard deviation and statistical analyzes of biological replicates.

Minor comments:

1. I consider that legends should not have a description of the results. For example, in line 781 (Legend Figure 1), I consider that the sentence “We observed  minor shifts in FTIR peaks of CDF and CDFHCD after inclusion complex formation indicating the successful formation of the CDFHCD inclusion complex.” should be removed since this should be described in Results section.

2. Also, in legends figures where says CDFHCD should say CDF-HCD.

3. The title of legends from figure 1 and 2 are the same. I would be better to modify the titles to remark which information is giving figure 1 and which information provide figure 2. For example, the title of Figure 2 could be: Morphological characterization of CDF-HCD complex and the interactions of the CDF within the cyclodextrin cavity.

4. In line 216, the sentence “The IC50 values were ranging from 0.81 to 1.05 μM in the case of CDF and 0.45-0.58 μM for different pancreatic cancer cell lines (Figure 3A-B)”, should be corrected because don´t say that  0.45-0.58 μM  range is for CDF-HCD complex. For example: “The IC50 values were ranging from 0.81 to 1.05 μM in the case of CDF and 0.45-0.58 μM in the case CDF-HCD complex for different pancreatic cancer cell lines (Figure 3A-B)”.

Author Response

In “Anticancer Activity of Novel Difluorinated Curcumin analog and its Inclusion Complex with 2-Hydroxypropyl-β-Cyclodextrin against Pancreatic Cancer”, Bhattacharyya et al., developed and characterized the inclusion complex of CDF with 2-hydroxypropyl-β-cyclodextrin which exerted greater antiproliferative effects against PDAC cell lines compared to CDF. They demonstrated that, both CDF and CDF-HCD, inhibited colony and spheroid formation, induced cell cycle arrest and apoptosis in PDAC cell lines. Interestingly, HCD improved hydrolytic stability of CDF.

Response: We are thankful to the reviewer for carefully analyzing our manuscript.

Major comments:

Comment  1. IC50 determination for proliferation was performed using 0 to 2.5 uM doses according to the figures. First, the legend figure say that cells were treated with 0 to 3 uM. Second, the table in Figure 3B show an IC50 of 2.5uM, but for accurate determination a curve with higher doses should be performed. Third, the authors should include error bars in the graphics. Also, for some cell lines, an IC50 higher to 5uM is indicated, this is not correct since the curve does not include this value. Also, the authors should analyse if the are significative differences in the IC50 of CDF-HCD compared to CDF. Finally, Materials and Methods should include how was the IC50 determined and IC50 in the table should include the mean and standard deviation.

Response: Per the reviewer’s suggestion, we added error bars and modified the graph showing cell viability to five micromolar concentrations. We also changed the writeup correcting the exact concentrations. The revised manuscript shows IC50 values in the table with standard deviation. Please refer to section 4.11 and Figure 3A-B.

Comment 2. When analysed CDF—HCD effect on cell cycle arrest the authors explain that treatment of CDF and CDF-HCD caused G2/M cell cycle arrest at 24 hours, while at 48hs an increase in cell number in the sub-G0 phase was observed. However, the histograms in Figure 5 shows a marked increase in sub-G0 cells after treatments at both times, 24 and 48hs. Also, in addition with the histograms of one representative experiment, authors should include a graphbar showing the mean and standard deviation of the biological replicates and also the statistical analyses and significance.

Response: We have corrected the write-up. The treatment with CDF and CDFHCD increased cell number in the sub-G0 phase after 24 and 48 h treatment in both MiaPaCa-2 and Panc-1 cells. The % cell number graph with statistics is now added to the manuscript. Please refer Figure 5A-B, Supplementary Figure 2A-B

Comment 3. Also, in Figure 5C authors should include the quantitation of the western blot, and it would be even better to include a bar graph showing mean and stardard deviation of the biological replicates.

Response: We included the quantification of all western blots in the revised manuscript. Please refer to Figure 5D.  

Comment 4. When analyze apoptosis, authors should include, in addition with the representatives dot-plots, a bar graph depicting mean, standard deviation and statistical analyses. Also, authors claims that “We saw an increased cell number over a period of 48 h in the early and late apoptosis stage after CDF or CDF-254 HCD treatment, when compared to control cells (Figure. 6A).”. They don’t describe the results at 24hs, that seem to be similar. Evenmore, how is autofluorescence in treated cells, it seems like curcumine interfere with annexin-FITC, since a shift to the right is observed in double negative cell population. Hence, the effect in late apoptosis is clear, but I’m not sure about the percentage indicated to early apoptotic cells.

Response: This is an excellent suggestion. We added the graph showing quantification and statistics in the manuscript. To address the comment about autofluorescence, we have repeated the experiments with negative control of CDF and CDFHCD. We observed 5-8% of cells in the early apoptotic phase, while there was no interference in the late apoptosis or necrosis phase. We modified the write-up accordingly. Please refer to section 2.7.3, Figure 6A, Supplementary Figure 3A.

Comment 5. In Figure 6C, density quantitation of each band should be showed and it would be better to include a bar graph showing mean, standard deviation and statistical analyzes of biological replicates.

Response: We included the quantification of all western blots in the revised manuscript. Please refer to Supplementary Figure 3B.

Comment 6. In Figure 7C, density quantitation of each band should be showed and it would be better to include a bar graph showing mean, standard deviation and statistical analyzes of biological replicates.

Response: We included the quantification of all western blots in the revised manuscript. Please refer to Supplementary Figure 4.

Minor comments:

Comment 7. I consider that legends should not have a description of the results. For example, in line 781 (Legend Figure 1), I consider that the sentence “We observed  minor shifts in FTIR peaks of CDF and CDFHCD after inclusion complex formation indicating the successful formation of the CDFHCD inclusion complex.” should be removed since this should be described in Results section. 

Response: We have modified the figure legends per the reviewer’s suggestion in the revised manuscript.

Comment 8. Also, in legends figures where says CDFHCD should say CDF-HCD. 

Response: We changed the label per the reviewer’s suggestion in the revised manuscript.

Comment 9. The title of legends from figure 1 and 2 are the same. I would be better to modify the titles to remark which information is giving figure 1 and which information provide figure 2. For example, the title of Figure 2 could be: Morphological characterization of CDF-HCD complex and the interactions of the CDF within the cyclodextrin cavity. 

Response: We have modified the legends per the reviewer’s suggestion in the revised manuscript.

Comment 10. In line 216, the sentence “The IC50 values were ranging from 0.81 to 1.05 μM in the case of CDF and 0.45-0.58 μM for different pancreatic cancer cell lines (Figure 3A-B)”, should be corrected because don´t say that  0.45-0.58 μM  range is for CDF-HCD complex. For example: “The IC50 values were ranging from 0.81 to 1.05 μM in the case of CDF and 0.45-0.58 μM in the case CDF-HCD complex for different pancreatic cancer cell lines (Figure 3A-B)”.

Response: We thank you for noticing this. We corrected this in the revised manuscript.

Reviewer 3 Report

Review Bhattacharya et al 2023

I read with a great interest the article by Bhattacharyya et al entitled “anticancer activity of novel difluorinated curcumin analog and its inclusion complex with 2-hydroxypropoyl-b-cyclodextrin (HCD) against pancreatic cancer” in which the authors show that HCD displays a higher water solubility and hydrolytic stability that were characterized by biophysics procedures. They also demonstrate that CDF and CDF-HCD inhibit pancreatic cancer proliferation. The manuscript is well written and easy to follow. As a non specialist of biophysics measurements, I was not able to eluate the quality and the interpretation of bio physic experiments. The discoveries on pancreatic cancer cell proliferation are promosing. However, I do not think that the manuscript should be accepted in the present form in IJMS journal, too many experiments lack statistical validation and have been performed only once See my explanations below:

Major points:

1- To my opinion, the introduction section is too long and should shortened for a better reading of the manuscript.

2- In figure 3A, it is not mentioned in the figure legend whether the experiments have been performed in triplicate or it is a single experiment. If so, please indicate the figure the SEM for each time point and a statistical p value.

3- In Suppl Fig S2, the authors test the toxicity of CDF-HCD on hPNE cells and claim that CDF-HCD has no cytotoxicity on normal cells. Once again, there is no SEM value on the graph. More importantly, only one cell line is tested which is weak to generalize. Moreover, hPNE cells are not normal but immortalized. I think that the data are not strong enough to claim an absence of toxicity on normal cells in general. This statement must be removed.

4- In Figure legends 4-5-6-7 and 8, the authors add “results” which have nothing to do in a figure legend that should only describe the experimental procedures and not the obtained results.

5- In Figure 5A and Suppl Table 1, once again, no statistical values are indicated. I do not know whether the experiment has been performed at least in triplicate. If it is the case, the SEM should be indicated.

6- In Figure 5C, the authors claim CDF-HCD induce a cell cycle arrest by inhibiting Cyclin D1 which prevents proliferation. First of all, there is no quantification of the Westenr blot analysis. More importantly, Cyclin D1 is a known cell cycle regulated protein. It is likely that CDF-HCD blocks the cell in a specific cell cycle phase. The consequences will be an increase of Cyclin D1 expression. Cyclin D1 increased expression would not be a cause of cell cycle arrest but rather a consequence. This statement should be removed of the manuscript.

Figure 6A, C , 7C, Suppl Figure S1: no statistical values, no quantifications of Western blot analyses

Minor points:

Page 3 line 84: curcumin has a “lower” water solubility…..compared to what ?  

Page 8 line 218: please indicate in the text that IC50 for CDF is at 72h

Page 13, line 365: indicate the way that the hPNE cells have been immortalized

Page 15, line 428: please indicate the dilution of your antibodies

Author Response

I read with a great interest the article by Bhattacharyya et al entitled “anticancer activity of novel difluorinated curcumin analog and its inclusion complex with 2-hydroxypropoyl-b-cyclodextrin (HCD) against pancreatic cancer” in which the authors show that HCD displays a higher water solubility and hydrolytic stability that were characterized by biophysics procedures. They also demonstrate that CDF and CDF-HCD inhibit pancreatic cancer proliferation. The manuscript is well written and easy to follow. As a non specialist of biophysics measurements, I was not able to eluate the quality and the interpretation of bio physic experiments. The discoveries on pancreatic cancer cell proliferation are promosing. However, I do not think that the manuscript should be accepted in the present form in IJMS journal, too many experiments lack statistical validation and have been performed only once See my explanations below: 

Comment 1. To my opinion, the introduction section is too long and should shortened for a better reading of the manuscript. 

Response: We totally agree with your point. However, several reports have been published on CDF's mechanism and anticancer activity. We thought this was a good opportunity to summarize it to ensure readers understand this curcumin derivative's importance.

Comment 2. In figure 3A, it is not mentioned in the figure legend whether the experiments have been performed in triplicate or it is a single experiment. If so, please indicate the figure the SEM for each time point and a statistical p value. 

Response: We thank the reviewer. All experiments are performed three times. We included all experiments' quantification (with SD) in the revised manuscript. Please refer to Figure 3A-B.  

Comment 3. In Suppl Fig S2, the authors test the toxicity of CDF-HCD on hPNE cells and claim that CDF-HCD has no cytotoxicity on normal cells. Once again, there is no SEM value on the graph. More importantly, only one cell line is tested which is weak to generalize. Moreover, hPNE cells are not normal but immortalized. I think that the data are not strong enough to claim an absence of toxicity on normal cells in general. This statement must be removed. 

Response: This is an excellent point. We added SD to the graph. Please refer to Supplementary Figure 1. We agree with the reviewer’s point and have deleted the statement accordingly in the revised manuscript. We modified the writeup about the absence of toxicity to normal cells.  

Comment 4. In Figure legends 4-5-6-7 and 8, the authors add “results” which have nothing to do in a figure legend that should only describe the experimental procedures and not the obtained results. 

Response: We modified the figure legends in the revised manuscript.

Comment 5. In Figure 5A and Suppl Table 1, once again, no statistical values are indicated. I do not know whether the experiment has been performed at least in triplicate. If it is the case, the SEM should be indicated.

Response: We included the quantification of all experiments, including western blots, in the revised manuscript. Please refer to Figure 5, Supplementary Figures 2A-B, 3A-B, Supplementary Figure 4

Comment 6. In Figure 5C, the authors claim CDF-HCD induce a cell cycle arrest by inhibiting Cyclin D1 which prevents proliferation. First of all, there is no quantification of the Westenr blot analysis. More importantly, Cyclin D1 is a known cell cycle regulated protein. It is likely that CDF-HCD blocks the cell in a specific cell cycle phase. The consequences will be an increase of Cyclin D1 expression. Cyclin D1 increased expression would not be a cause of cell cycle arrest but rather a consequence. This statement should be removed of the manuscript. 

Response: We agree with the reviewer’s assessment. We included the Cyclin D1 western blot quantification and removed the statement in question from the revised manuscript. We observed decreased levels of cyclin D1. We deleted the suggested statement. Please refer to Figure 5C-D

Comment 7. Figure 6A, C , 7C, Suppl Figure S1: no statistical values, no quantifications of Western blot analyses

Response: We included the quantification of all western blots in the revised manuscript. Please refer to Supplementary Figure 3A-B, Supplementary Figure 4

Minor points:

Comment 8. Page 3, line 84: curcumin has a “lower” water solubility…..compared to what ?  

Response: There is no comparison. Curcumin generally has low solubility, limiting its clinical use because of its low bioavailability.

Comment 9. Page 8 line 218: please indicate in the text that IC50 for CDF is at 72h.

Response: We have modified the sentence as per your suggestion.

Page 13, line 365: indicate the way that the hPNE cells have been immortalized.

Response: We did not immortalize the HPNE cell line in our lab. This standard cell line is available from ATCC (cat no. CRL-4023). We cultured the cell line per the guidelines of ATCC.

Comment 10. Page 15, line 428: please indicate the dilution of your antibodies

Response: We have used antibody dilutions as per the manufacturer’s instructions. We have stated the dilutions in the material methods in the revised manuscript.

Round 2

Reviewer 2 Report

In the second version of “Anticancer Activity of Novel Difluorinated Curcumin analog and its Inclusion Complex with 2-Hydroxypropyl-β-Cyclodextrin against Pancreatic Cancer”, Bhattacharyya et al have improved their manuscript. All the comments were properly answered. I think that this manuscript should be published in the present form.

Author Response

We are thankful to the reviewer for recommending our article for acceptance.